# Combinations of Hydrogels and Mesenchymal Stromal Cells (MSCs) for Cartilage Tissue Engineering—A Review of the Literature

**DOI:** 10.3390/gels7040217

**Published:** 2021-11-16

**Authors:** Mike Wagenbrenner, Susanne Mayer-Wagner, Maximilian Rudert, Boris Michael Holzapfel, Manuel Weissenberger

**Affiliations:** 1Department of Orthopaedics and Trauma Surgery, Musculoskeletal University Center Munich (MUM), University Hospital, LMU Munich, Marchioninistraße 15, 81377 Munich, Germany; mike.wagenbrenner@med.uni-muenchen.de (M.W.); susanne.mayer@med.uni-muenchen.de (S.M.-W.); m-rudert.klh@uni-wuerzburg.de (M.R.); direktion.opmr@med.uni-muenchen.de (B.M.H.); 2Department of Orthopaedic Surgery, University of Würzburg, König-Ludwig-Haus, Brettreichstr. 11, 97074 Würzburg, Germany

**Keywords:** hydrogels, osteoarthritis, cartilage defects, MSCs, cartilage regeneration, tissue engineering

## Abstract

Cartilage offers limited regenerative capacity. Cell-based approaches have emerged as a promising alternative in the treatment of cartilage defects and osteoarthritis. Due to their easy accessibility, abundancy, and chondrogenic potential mesenchymal stromal cells (MSCs) offer an attractive cell source. MSCs are often combined with natural or synthetic hydrogels providing tunable biocompatibility, biodegradability, and enhanced cell functionality. In this review, we focused on the different advantages and disadvantages of various natural, synthetic, and modified hydrogels. We examined the different combinations of MSC-subpopulations and hydrogels used for cartilage engineering in preclinical and clinical studies and reviewed the effects of added growth factors or gene transfer on chondrogenesis in MSC-laden hydrogels. The aim of this review is to add to the understanding of the disadvantages and advantages of various combinations of MSC-subpopulations, growth factors, gene transfers, and hydrogels in cartilage engineering.

## 1. Introduction

Osteoarthritis (OA) affects more than 10% of men and 18% of women worldwide and places an enormous socio-economic burden on health care systems worldwide, with the number of joint replacement surgeries projected to increase steadily [1,2,3]. OA is characterized by traumatic or degenerative lesions to hyaline cartilage, which is a highly specialized, avascular, and brady trophic tissue covering the surface of diarthrodial joints [4]. As a result, damage to hyaline cartilage or osteochondral lesions naturally results in lasting defects or the formation of inferior fibrocartilage, which is why surgical and regenerative treatment methods for cartilage repair have gained growing interest [4,5]. 

Tissue engineering combines the use of growth factors, gene transfer, and biomaterials to optimize chondrogenic differentiation and maintenance of a chondrogenic phenotype in seeded cells. 

Cell-based approaches, such as tissue engineering (TE), combine the use of chondrogenic growth factors, cells, and functional scaffolds to further optimize the treatment of cartilage defects (Figure 1) [6]. Multipotent mesenchymal stromal cells (MSCs) have emerged as a promising cell source for use in cartilage engineering [7,8,9]. MSCs carry a characteristic set of surface markers, grow plastic adherent, can be differentiated toward the osteogenic, adipogenic, and chondrogenic lineage in vitro, and have been shown to reside in various, easily accessible adult and human fetal tissues [10,11]. Multiple studies have proven the potential of MSCs and their secretome to promote the natural healing and new formation of cartilage tissue in vitro and in vivo [7,12]. Despite good in vitro and in vivo data, limitations to the use of MSCs include loss of transplanted cells upon transplantation, insufficient chondrogenic differentiation, osteogenic de-differentiation, chondrogenic hypertrophy, or failed integration in targeted defects [4,13]. 

Therefore, to optimize clinical approaches for cell-based cartilage engineering, a beneficial 3D-microenvironment is necessary, containing a specific combination of biomaterials and growth factors to further enhance chondrogenesis in seeded cells. These biomaterials include hydrogels which are formed from various natural, synthetic, or modified polymers, retain large amounts of water and mimic the natural structure of hyaline cartilage to optimize chondrogenic differentiation and enhance cell functionality in MSCs [14]. Although the lack of mechanical stability, controlled biodegradability, or immunogenicity pose important challenges, hydrogels have been approved and successfully used in clinical approaches for the engineering of cartilage or intervertebral discs [12,15,16,17]. Cell-laden hydrogels can be manufactured according to defect composition, can be applied via minimal-invasive approaches, and may promote the repair of full-thickness cartilage defects [18]. In addition, the combination of natural and synthetic hydrogels and growth factors has been shown to enhance chondrogenic differentiation, maintain chondrogenic phenotype, and limit chondrogenic hypertrophy in seeded MSCs [14,15,19]. 

To further contribute to the field of cartilage engineering, we compared the use of MSCs and natural, synthetic, or modified hydrogels in this narrative review. Different types of hydrogels and subpopulations of MSCs were discussed and examined regarding specific advantages and disadvantages. Finally, we compared the combination of different MSCs, hydrogels, growth factors, or gene transfer for cartilage engineering to examine which composition offers the best results according to the current scientific literature. 

## 2. Results

### 2.1. Hydrogels in Cartilage Tissue Engineering

Hydrogels consist of three-dimensional networks built from hydrophilic, polymeric biomaterials that are crosslinked through either covalent or physical bindings [20]. Current research focuses on the use of these three-dimensional hydrogels, which mimic the extracellular matrix (ECM) of hyaline cartilage to further optimize the treatment of cartilage defects. Both cell-free and cell-laden hydrogels have been used to treat cartilage defects [14,21,22,23,24]. Hydrogels used for the treatment of cartilage defects have to be highly biodegradable and biocompatible, possess strong water binding capacity enabling them to double their size through swelling, and present a certain porosity [20]. In addition, injectable hydrogels offer the advantage that they can be shaped to fit into individual-sized and -shaped cartilage defects [14,25]. Hydrogels can be divided into natural, synthetic, or modified natural hydrogels depending on the polymeric material they are built from. Natural hydrogels can be further separated into polysaccharide-based hydrogels formed from agarose (AG), alginate (AL), glycosaminoglycans (GAGs), and chitosan (CH), as well as protein-based hydrogels formed from collagen (COL), elastin (EL), gelatin (GEL), or other polymers [26]. Synthetic hydrogels are formed from polymers, such as polyethylene glycol (PEG), polyvinyl alcohol (PVA), or Poly(lactide-co-glycolide) (PLGA) [5,20,25,27]. In an approach to combine the benefits of both natural and synthetic hydrogels in the field of cartilage TE, modified hydrogels combining multiple polymers have been used [26,28]. 

### 2.2. Polysaccharide-Based Hydrogels in Cartilage Tissue Engineering

GAGs can act as the base for polysaccharide-based hydrogels. An example for GAGs is hyaluronic acid (HA) which forms the most abundant component in the ECM of human hyaline cartilage and is also present in the ECM of other mammalian connective tissues where it mostly acts as a lubricant [26,29]. HA also contributes toward the resistance of hyaline cartilage toward shear and compressive forces. As a polymer, HA is strongly hydrophilic and highly biodegradable, possesses low adhesiveness, and provides a microenvironment similar to natural hyaline cartilage offering great conditions for the use in TE [26,28]. However, HA hydrogels can be hydrolyzed and therefore are unstable and easily degradable at body temperature. 

Further, HA exhibits natural surface antigens that influence metabolism, inflammation, and proliferation in seeded chondrocytes [30]. 

HA hydrogels have been shown to increase the expression of chondrogenic marker genes and the synthesis of chondrogenic marker molecules in seeded MSCs and chondrocytes [8,31,32,33]. Hydrogels made from HA promoted early chondrogenesis in seeded MSCs by enhancing the synthesis of aggrecan and collagen type II, which is considered the gold standard for successful chondrogenic differentiation of MSCs in vitro. Hence, cell-laden HA hydrogels have been studied to form neocartilage tissue in vitro and in vivo [33,34]. 

To overcome the poor stability and control of biodegradability of HA hydrogels, they have been subject to various modifications, such as esterification of hydroxyl or carboxyl groups [35]. In addition, hydrophobic modification to hydrophilic HA can be made with polylactic acid (PLA) or ammine to create self-assembling and more modifiable, stable hydrogels [36]. Natural co-polymers derived from HA examined by Oldinski et al. showed modifiable viscoelasticity and porosity as well as biocompatibility and could be used for the treatment of osteochondral defects when used together with MSCs [6]. Further chondrocytes seeded in HA hydrogels modified with elastin-like protein showed enhanced expression of chondrogenic marker genes and GAG deposition with rising concentrations of HA [37]. Conjugation of sulfate groups to HA hydrogels allowed the slowing down biodegradation and led to a retention of seeded growth factors, thus enhancing chondrogenesis and preventing hypertrophic de-differentiation in encapsuled MSCs both in vitro and in vivo in animal OA models [38]. 

The natural polymer AL is a major component of cell walls in brown algae and capsules of certain bacteria [26]. The structure and mechanical properties of AL depend on the deposition of both monomers it is formed from [26]. AL can be physically crosslinked with divalent cations at room temperature, making it moldable and useful in the field of 3D bioprinting [39]. AL hydrogels also offer low costs and cytotoxicity, high biocompatibility, low immunogenicity, and low inflammatory characteristics leading to their frequent use in various biomedical applications [40]. However, AL hydrogels present low biodegradability despite insufficient mechanical stability in vivo and offer poor cell-adhesive properties [14,26,41]. 

MSC-laden AL hydrogels have been shown to promote the formation of repair tissue and regeneration of osteochondral defects in rabbit models [42]. Research has also shown that hypoxia mimicking AL hydrogels loaded with growth factors guide seeded cells toward a more chondrogenic phenotype while preventing osteogenic and hypertrophic de-differentiation [43]. This effect may be enhanced when paired with shear and compression forces [42]. The combination of HA hydrogels with AL microspheres has been shown to retain the activity of transported growth factors and induce chondrogenesis in encapsulated MSCs both in vitro and in vivo [34]. 

In the field of cartilage TE, AL hydrogels have been linked with sulfate groups to promote a more functional tissue supporting cell growth and proliferation while enhancing the synthesis of cartilage-specific matrix proteins, such as collagen type II, in seeded chondrocytes [44]. 

AG offers great biocompatibility, water-solubility, is non-immunogenic, and possesses adjustable mechanical characteristics making it a widely used natural polymer in hydrogels for TE [45]. In addition, AG possesses thermally-reversible gelatin characteristics, is stable at body temperature, and soluble at temperatures over 65 °C—great conditions for the use in the minimal-invasive treatment of cartilage defects with the use of cell-laden hydrogels. Further, AG hydrogels provide a great balance between viscoelastic properties and stiffness with an adjustable water-binding capacity [14,45]. 

Research has shown that AG hydrogels support chondrogenic differentiation, synthesis of chondrogenic ECM, and the maintenance of chondrogenic phenotype in seeded cells when used for cartilage TE in vitro and in vivo [46,47,48]. However, AG may lead to less enhanced cell functionality, chondrogenic differentiation, and synthesis of cartilage-specific ECM components when compared to other polymers used in natural hydrogels for cartilage TE [14,49]. 

CH is a polymer derived from chitin which is formed from D-glucosamine and N-acetyl-D-glucosamine and acts as the major component of the exoskeleton in various arthropods [26]. CH is cost-effective, acts bacteriostatic, is highly biodegradable and biocompatible, and has structural similarities to GAGs found in the ECM of hyaline cartilage, which contribute to the resistance of hyaline cartilage toward shear and compression forces [14,26]. 

CH hydrogels have been shown to promote chondrogenesis and maintenance of chondrogenic phenotype in seeded cells as well as the deposition of cartilage-specific ECM components [50,51]. Sheeshy et al. showed that CH hydrogels may promote and maintain a superior chondrogenic phenotype while limiting hypertrophic de-differentiation in seeded MSCs in comparison to other natural polymers, such as AL or fibrin [48]. However, CH offers poor mechanical properties and is sensitive to temperature and pH changes, although some of these limitations can be overcome by modifications made to CH hydrogels [26,52]. 

### 2.3. Protein-Based Hydrogels in Cartilage Tissue Engineering

COL—especially COL type II—is the most abundant structural protein present in the natural ECM of hyaline cartilage. In addition, COL type II is known as the gold marker for successful chondrogenesis in MSCs in vitro [26,53]. However, hydrogels used for cartilage TE are often based on COL type I due to its high biocompatibility, vast safety, and clinical approval [53]. COL-based scaffolds also enhance cell functionality, phenotype maintenance, and cell proliferation upon the binding of cell receptors to natural ligands [53,54]. 

We and others showed that COL type I hydrogels enhance chondrogenic differentiation and maintenance of chondrogenic phenotype in seeded MSCs in vitro, especially when paired with specific growth factors or gene transfer [19,53,55]. However, research has shown that both COL I-based scaffolds promote chondrogenesis in seeded MSCs with a possible superior synthesis of cartilage-specific ECM components but no significant differences in the expression of chondrogenic marker genes [55,56,57]. 

Major disadvantages when using COL type II hydrogels are their possible arthritogenic potential in combination with low clinical approval [53]. Limitations for pure COL type I-based hydrogels include limited mechanical properties, shrinkage, and limited induction of chondrogenic differentiation in seeded cells [53]. In addition, chondrocytes seeded in COL type I hydrogels underwent de-differentiation presented by decreased expression of chondrogenic marker genes, which could be due to contraction of COL at low concentrations promoting chondrocyte condensation [53,58]. 

GEL is a denatured form of COL produced by hydrolyzation [47]. GEL offers many advantages seen in COL hydrogels, such as high biocompatibility and biodegradability, as well as material–cell interaction enhancing cell functionality, chondrogenesis, and phenotype maintenance in seeded cells while showing better mechanic stability [59]. However, crosslinks in GEL hydrogels may offer inferior stability at body temperature requiring further modification with other polymers [60]. Hydrogels bioprinted from gelatin-methacryloyl (gelMA) and gellan gum have been shown to promote the production of cartilage-specific ECM components in seeded chondrocytes in vitro [61]. 

Silk fibroin is a new polymer used to form protein-based hydrogels for cartilage TE. Although silk fibroin hydrogels promote the synthesis of matrix proteins similar to the ECM in hyaline cartilage when loaded with chondrocytes and MSCs, this induction of chondrogenesis in seeded cells may be inferior to that seen in other natural hydrogels [14,48]. 

### 2.4. Synthetic Hydrogels in Cartilage Tissue Engineering

Synthetic hydrogels are based on industrially manufactured polymers making them highly adjustable regarding porosity, biocompatibility, and biodegradability, as well as mechanically strong and reproducible. Research has shown that synthetic hydrogels alone or natural hydrogels modified with synthetic components may further improve cartilage TE [15]. However, synthetic polymers remain biologically inert, limiting their influence on cell functionality, as well as cell adherence, and are relatively expensive. 

Due to its clinical approval, PEG remains a very popular synthetic polymer used for hydrogel formation. In addition, PEG can be easily modified while possessing great mechanical properties [15]. Chondrocytes embedded in PEG-HA hydrogels maintained their chondrogenic phenotype, showed increased functionality and limited hypertrophy [62]. PEG hydrogels modified with chondroitin sulfate may further limit hypertrophy during chondrogenesis in seeded MSCs [62]. Further, PEG-diacrylate (PEGDA) hydrogels have been shown to promote the formation of cartilage-specific ECM alone and in combination with seeded MSCs in vitro and have led to promising short-term results when used for the clinical treatment of cartilage defects in vivo [22]. In addition, PEGDA hydrogels modified with fibrinogen have been shown to enhance chondrogenic differentiation while limiting hypertrophic de-differentiation in seeded MSCs [63]. 

PLGA hydrogels modified with fibrin have been shown to promote the repair of full-thickness cartilage defects in rabbits when paired with MSCs and chondrogenic growth factors [64]. PVA also offers great properties for emulsification and cell adhesion and mechanical stability and performance similar to that of natural hyaline cartilage [65]. In addition, modified PVA-CH hydrogels promoted chondrogenic differentiation and the synthesis of cartilage-specific ECM components in seeded MSCs in vitro [66]. Poly(N-vinylcaprolactam) (PVCL) has also been used as a thermosensitive polymer in hydrogels for cartilage TE. Seeding cells in PVCL hydrogels led to successful chondrogenic differentiation accompanied by deposition of cartilage-specific ECM components as well as high cell viability [15]. 

### 2.5. Comparison of Hydrogels in Cartilage Tissue Engineering

In summary, natural hydrogels offer great properties for biocompatibility, biodegradability, cell viability, and the promotion of cell functionality by cell-material interactions (Table 1). Major disadvantages include low mechanical stability and high variability. In contrast, synthetic hydrogels offer highly tunable characteristics, such as porosity, viscoelasticity, and biodegradability. The specific advantages and disadvantages for protein- and polysaccharide-based hydrogels, as well as synthetic hydrogels, are listed in Table 1.

Other than PEG, synthetic polymers are mostly not clinically approved, limiting their application in the field of cartilage TE. When comparing natural polymers for the fabrication of hydrogels, polysaccharide-based hydrogels based on HA and AG as well as protein-based COL hydrogels possess good mechanical stability, especially when modified with other natural and synthetic polymers, enhance cell functionality and differentiation by material–cell interaction, and offer good biocompatibility and biodegradability. All these natural hydrogels have shown promising results when used for cartilage engineering and repair both in vitro and in vivo [14,33,45,53]. Still, more research regarding newer natural and synthetic polymers as well as modified hydrogels is necessary to further determine the optimal hydrogel for cartilage TE. 

### 2.6. MSC-Laden Hydrogels for Cartilage Tissue Engineering

Different kinds of cells and growth factors have been used to further enhance the pro-chondrogenic effects of hydrogels in the field of cartilage TE. Besides chondrocytes, multipotent or even pluripotent cells used in the field of cartilage TE include embryonic stem cells (ESCs), induced pluripotent stem cells (iPSCs), and different subpopulations of MSCs [16,67]. Interestingly, research has shown that different subpopulations of MSCs also possess different chondrogenic differentiation potential and vary in their set of surface antigens depending on the source of tissue they originate from [68,69,70]. In contrast to the pluripotent ESCs and iPSCs, multipotent MSCs raise fewer ethical concerns and can be isolated in large numbers from almost all vascularized adult and fetal tissues using a minimal invasive surgical approach. Therefore, MSCs have emerged as a promising cell source for cartilage TE. However, there remains disagreement regarding the optimal combination of MSCs and hydrogels as well as growth factors which is why this review focuses on recent advances in the combinations of these variables to further optimize cartilage TE (Figure 2). 

### 2.7. MSCs in Cartilage Tissue Engineering

The minimal criteria for MSCs, as defined by the International Society for Cell & Gene Therapy (ISCT), state that MSCs are plastic-adherent cells, presenting a characteristic set of surface antigens that possess adipogenic, osteogenic, and chondrogenic differentiation capacity [71]. MSCs present a promising cell source in the field of TE since they can be isolated in abundance from a wide variety of fetal and adult tissue sources and multiplied by cultivation in vitro. In addition, MSCs possess low immunogenicity and raise few ethical concerns. Although bone marrow-derived MSCs (BMSCs) are still viewed as the gold standard, synovial-derived or adipose tissue-derived MSCs may possess superior chondrogenic differentiation capacity while offering easier access to native tissues [70,72]. While chondrogenesis in MSCs can be stimulated by combination with scaffolds and growth factors, research has revealed that their secretome may also positively influence local tissue and cartilage repair. 

### 2.8. BMSC-Laden Hydrogels for Cartilage Tissue Engineering

The most common combination of MSCs and hydrogels used for in vivo cartilage engineering in different animal models are BMSCs combined with HA hydrogels (Table 2) [23,73,74,75,76,77,78,79,80,81]. Multiple studies used BMSC-laden HA hydrogels for the treatment of osteochondral defects in animal models. Kim et al. combined injectable HA hydrogels with BMSCs for the treatment of osteochondral defects. They found that the combined use of HA and BMSCs led to a significantly enhanced healing of osteochondral defects, especially when using multiple HA injections [23]. Lee et al. showed similar results when treating osteochondral defects in pigs with superior macroscopic and histological regeneration of hyaline cartilage when combining the use of HA injections and BMSCs in comparison to HA injections or negative controls [75]. Another study examined the effects of BMSC-laden HA hydrogels in the treatment of osteochondral defects of the femorotibial joint in horses. Results showed that microfracture combined with BMSC-laden HA hydrogels led to superior tissue quality in repair tissue compared to the treatment with microfracture and HA hydrogels alone, while no clinically significant differences were observed [73]. Saw et al. found similar results when treating osteochondral defects in goats with superior results in histological hyaline cartilage repair when combining microfracture, HA injections, and BMSCs compared to the use of HA injections and microfracture alone [74]. A clinical trial examined the use of BMSC-laden HA hydrogels for the treatment of osteochondral defects of the knee and found significant improvements in functionality and pain reduction compared to patients treated with microfracture alone (Table 3) [82]. 

Other researchers combined BMSCs and hydrogels for the treatment of OA in different animal models. Chiang et al. found that BMSCs enhance the positive effects of HA hydrogels when treating OA in rabbit models. The combined use of BMSC and HA hydrogels led to less cartilage loss, fewer surface abrasions, and significant improvements in histological scores and cartilage content compared to using HA hydrogels alone [77]. Other studies confirmed the positive effects on clinical, radiographic, or histological outcomes in animal models of OA when combining BMSCs and HA hydrogels in comparison to negative controls or the use of BMSCs or HA hydrogels alone [14,79,80]. Further, DeSando et al. stated that HA supports cell migration to hyaline cartilage when using both BMSCs or bone marrow concentrate, with superior results in the treatment of OA in rabbits when using bone marrow concentrate and HA hydrogels [76]. In contrast, another study showed that the separate use of BMSCs and HA hydrogels may be beneficial for the treatment of OA in rats [78].

However, BMSCs have also been combined with various other natural or synthetic hydrogels in cartilage TE. Pascual-Garrido et al. treated critical-sized chondral defects in rabbit knees with BMSC-laden photopolymerizable hydrogels and found higher scores after histological cartilage examination in comparison to untreated defects or those only treated with hydrogels [83]. Another study used BMSC-laden AL hydrogels for the treatment of osteochondral defects in the knee of rabbits and found that chondral repair tissue showed more hyaline cartilage-like properties when compared to untreated groups [42]. Choi et al. combined BMSCs treated with resveratrol and gelatin-based hydrogels for the treatment of osteochondral defects in rabbit knees. They found that treatment with these cell-laden hydrogels led to the formation of hyaline cartilage-like repair tissue with vast amounts of collagen type II and increased GAG deposition compared to treatment solely with hydrogels or untreated MSC-laden hydrogels [84]. In addition, Kim et al. examined the therapeutic effects of self-assembled peptide (SAP) hydrogels and BMSCs on OA in rats. The authors found that the combined use of SAP hydrogels and BMSCs led to anti-inflammatory effects, decreased levels of apoptosis biomarkers as well as chondroprotective effects on a histological level [85]. 

### 2.9. Hydrogels Combined with Other MSC-Subpopulations

Different MSC-subpopulations other than BMSCs may offer advantages regarding tissue accessibility, cell abundance, or chondrogenic differentiation capacity [70]. Therefore, multiple studies have focused on the combined use of different MSC-subpopulations with hydrogels (Table 4). Adipose-derived (AD)MSCs represent the second most used MSC subpopulation used in cartilage TE. One study examined the combined effects of ADMSCs and HA hydrogels on OA progression in sheep. Results showed decreased OA progression and increased cartilage regeneration efficacy in comparison to untreated groups or those treated with stromal vascular fraction and HA hydrogels [88]. Feng et al. found similar results combined with anti-inflammatory effects when combining the use of ADMSCs and HA hydrogels for the treatment of OA in sheep models [89]. A recent study by Sevastianov et al. pointed out that ADMSCs seeded in decellularized porcine articular cartilage may produce more cartilage-specific matrix proteins in comparison to MSCs treated in COL hydrogels in vitro [90]. In contrast, ADMSCs seeded in COL hydrogels led to improved cartilage repair in OA in rabbit models in vivo when compared to MSCs seeded in decellularized porcine articular cartilage [90]. A recent clinical trial evaluated the combined effects of ADMSC-laden polyglucosamine/glucosamine carbonate hydrogels and microfracture on osteochondral defects of the knee in forty-six patients. The results showed higher patient satisfaction and superior histological formation of hyaline cartilage in comparison to the control group, which was treated with microfracture alone (Table 3) [87].

Jia et al. seeded synovial fluid-derived MSCs in injectable CH hydrogels for the treatment of femoral full-thickness cartilage defects in rabbits [91]. Results revealed that MSC-laden hydrogels led to superior histological cartilage repair when compared to only hydrogels or controls. Wu et al. found that the combined use of HA hydrogels and human umbilical cord-derived MSCs may be an effective treatment for OA in minipig models [92]. Li et al. showed that the therapeutic use of arthroscopic flushing fluid-derived MSCs encapsulated in a polyPEGDA/HA hydrogel led to the significant repair of full-thickness cartilage defects as well as macroscopic smooth cartilage in rats [93]. 

### 2.10. Gene Transfer and Growth Factors Combined with Hydrogels for Cartilage Tissue Engineering

Hypertrophic and osteogenic de-differentiation remains a major problem regarding the repair of chondral and osteochondral defects using MSCs and may result in the formation of hypertrophic cartilage tissue or osteophytes in vivo [4]. Although the combination of MSCs with specific natural, synthetic, or modified hydrogels has shown promising results for the repair of cartilage defects both in vitro and in vivo, the combination with specific growth factors may further enhance and guide the differentiation of encapsulated MSCs (Table 5) [14,94,95].

Different growth factors from the transforming growth factor (TGF)-ß superfamily have been used to enhance chondrogenic differentiation of MSCs encapsulated in hydrogels. PLGA hydrogels were modified with bone morphogenetic protein (BMP)-2 to provide better support for encapsulated BMSCs [96]. Synthetic hydrogels modified with HA and preloaded with TGF-ß3 led to enhanced chondrogenesis in seeded BMSCs and showed increased mechanical strength [97]. Further hypoxia mimicking AL hydrogels loaded with BMP-2, TGF-ß3 reduced hypertrophic de-differentiation in seeded BMSCs [43]. Platelet lysate contains various growth factors able to enhance the formation of a hyaline cartilage-like ECM when incorporated in MSC-laden hydrogels [98]. 

Further, we and others showed that gene transfer of different members of the TGF-ß superfamily led to enhanced chondrogenesis of BMSCs seeded in hydrogels in vitro and in vivo [19,99,100,101,103,104,105,106]. Xia et al. found that adenoviral gene transfer of TGF-ß1 (encoded by *TGFB1*) to BMSCs encapsulated in PGA hydrogels led to enhanced chondrogenesis in vivo [99]. Wang et al. showed that adenoviral gene transfer of TGF-ß3 (encoded by *TGFB3*) and BMP-2 (encoded by *BMP2*) to BMSCs encapsulated in demineralized bone matrix led to enhanced cartilage repair in pig models [100]. Different studies showed that the gene transfer of SRY-Box Transcription Factor (SOX)-9 (encoded by *SOX9*) may further enhance chondrogenesis in BMSC seeded in natural or synthetic hydrogels while limiting hypertrophic de-differentiation [19,102,107]. Cao et al. showed that adenoviral gene transfer of *SOX9* to BMSCs seeded in PGA hydrogels led to enhanced cartilage repair in rabbit models in comparison to conventional BMSC-laden PGA hydrogels [101]. Further, gene transfer of different members of the TGF-ß superfamily to ADMSCs seeded in natural or synthetic hydrogels led to successful cartilage repair and delayed progression of OA in rat models [103,104]. 

## 3. Discussion

Hyaline cartilage is a highly specialized and complex tissue and with limited regenerative potential. Although cell-based methods have emerged as a successful treatment option for OA and cartilage defects major obstacles, such as loss of transplanted cells, adequate chondrogenic differentiation, and phenotype maintenance, as well as chondrogenic hypertrophy and osteogenic de-differentiation or integration difficulties, remain [4,12]. To further optimize TE approaches for cartilage repair, research has focused on different combinations of scaffolds, MSC-subpopulations, and growth factors. This review focused on the use of MSC-laden hydrogels and possible modifications using growth factors or gene transfer for use in cartilage engineering.

Different natural, synthetic, and modified hydrogels possess specific advantages and disadvantages, considering their clinical use. However, the modification of hydrogels offers to combine advantages of both natural and synthetic polymers to optimize biocompatibility, mechanical structure, biodegradability, and induction of chondrogenesis in seeded cells [5,14]. Further, different natural and synthetic hydrogels have been approved for clinical use, making them highly attractive scaffolds for cell-based cartilage engineering [12,26]. 

Although autologous chondrocytes are used in many current cell-based cartilage repair methods, such as autologous chondrocyte implantation (ACI), MSCs can be isolated in great numbers from various easily accessible tissues. In addition, MSCs possess greater proliferation potential while offering great chondrogenic differentiation capacity [69,108]. When choosing the right MSC-subpopulations for cartilage TE, tissue accessibility, cell abundance, and chondrogenic differentiation capacity have to be considered. Natural polymers in hydrogels have been shown to influence cell functionality in both chondrocytes and MSCs promoting chondrogenesis and the formation of hyaline cartilage-like ECM as well as limiting hypertrophic de-differentiation, although these effects are limited by biodegrading of biomaterials [14,28]. 

The success of cell-laden hydrogels in cartilage TE depends on the maintenance of cell viability, the synthesis of ECM components, and tissue integration [17]. The addition of specific growth factors or gene transfer to encapsulated MSCs may further influence cell functionality and suitability for cartilage TE as well as all of the variables mentioned above. As pointed out earlier, numerous growth factors from the TGF-ß superfamily have been shown to promote chondrogenesis in seeded MSCs in hydrogels in vitro and in vivo. Further gene transfer of *SOX9* has been shown to promote not only chondrogenic differentiation in seeded MSCs but also limit hypertrophic differentiation in MSC-laden hydrogels in vitro [17,19]. The combination with bioactive substances and MSCs may help to overcome the hurdle of rapid degradation in natural hydrogels, which often takes place faster than hydrogels can be replaced by de novo ECM [17].

To determine the ideal composition of MSC-subpopulations, loaded growth factors, or gene transfer and hydrogels, further research and clinical trials are necessary, especially regarding the use of more complex hydrogels modified with specific growth factors or gene transfer. In this context, bioprinted biphasic or triphasic biomaterials mimicking the multizonal construction of the osteochondral interface have emerged as a promising alternative [109,110]. These scaffolds offer an osseous and a chondral layer as well as a potential third layer mimicking calcified cartilage [110]. Although preclinical and clinical studies show encouraging results, the combination of more phasic biomaterials with specific MSC-subpopulations and growth factors to recreate the zonal structure of native hyaline cartilage remains challenging, and until now, no effective strategy to reliably direct zonal differentiation in seeded cells has been developed [110,111]. 

## 4. Conclusions

MSC-laden hydrogels offer a promising approach for the treatment of cartilage defects. Currently, the combination of natural hydrogels with BMSCs offers the most clinical experience. However, other MSC-subpopulations may provide various advantages, including better accessibility of the native tissue source as well as the improved proliferative and chondrogenic potential of cells. In addition, modified hydrogels may improve chondrogenic differentiation, maintain chondrogenic phenotype and decrease hypertrophic differentiation in seeded cells while providing optimal biocompatibility, stability, and biodegradability. Bioprinted biphasic scaffolds are a promising option for the reconstruction of the functional osteochondral unit. 

## 5. Material and Methods

### 5.1. Methods Used for Literature Research

The research for this narrative review was performed with the PubMed database using the following search strings: “stem cells” OR “mesenchymal stromal cells” OR “MSCs” AND “hydrogel” AND “gene transfer” OR “growth factors” AND “cartilage repair” OR “cartilage defects”. We selected papers published in English that were released up to August 2021. We included preclinical and clinical studies examining the effects of MSC-laden hydrogels alone or in combination with chondrogenic growth factors for cartilage tissue engineering. We also included in vitro and preclinical studies examining the effects of MSC-laden hydrogels in combination with gene transfer for the regeneration of hyaline cartilage. 

### 5.2. Results of Literature Research

The results of our literature research are pictured in Figure 3. The described search string led to 357 initial articles. After excluding duplicates, 200 papers remained. Following this procedure, we performed a manual review based on displayed study titles and abstracts which was performed by three independent reviewers. We found that 32 articles represented reviews or presentations, and 45 studies did not cover the use of cell and cell-laden hydrogels for the treatment of cartilage defects. After a full review, out of the consisting articles, 30 papers did not relate to MSCs resulting in a total of 93 studies after screening. 

After excluding 22 articles with no availability of full-text, 3 studies with no available study protocol, and 38 studies that did not present a fitting study protocol after review of the abstract or materials and methods section, we included 27 preclinical studies and 3 clinical studies in this present narrative review discussing the engineering of cartilage defects with MSC-laden hydrogels. We divided the search results into preclinical research examining the effects of MSC-laden hydrogels, MSC-laden hydrogels in combination with growth factors, and MSC-laden hydrogels in combination with gene transfer, as well as clinical studies examining the effect of MSC-laden hydrogels on cartilage repair. 

## Figures and Tables

**Figure 1 gels-07-00217-f001:**
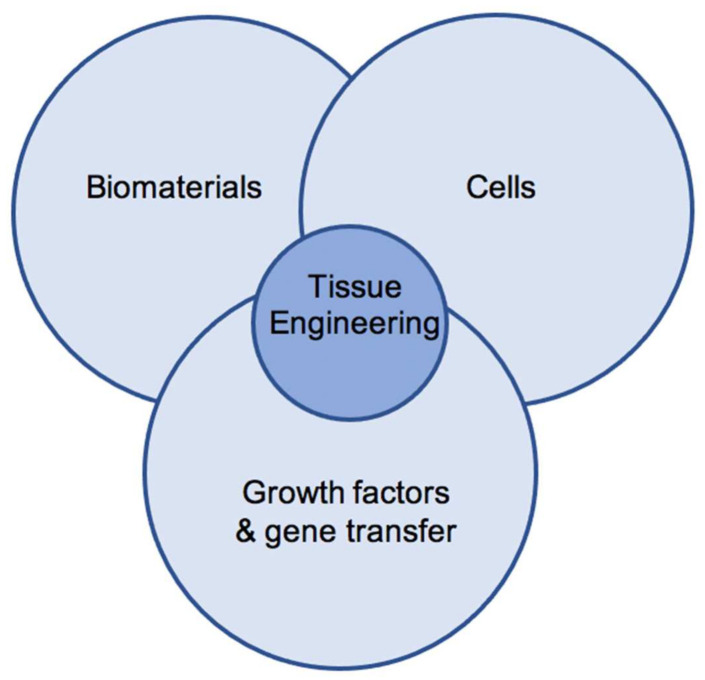
The triade of cartilage tissue engineering.

**Figure 2 gels-07-00217-f002:**
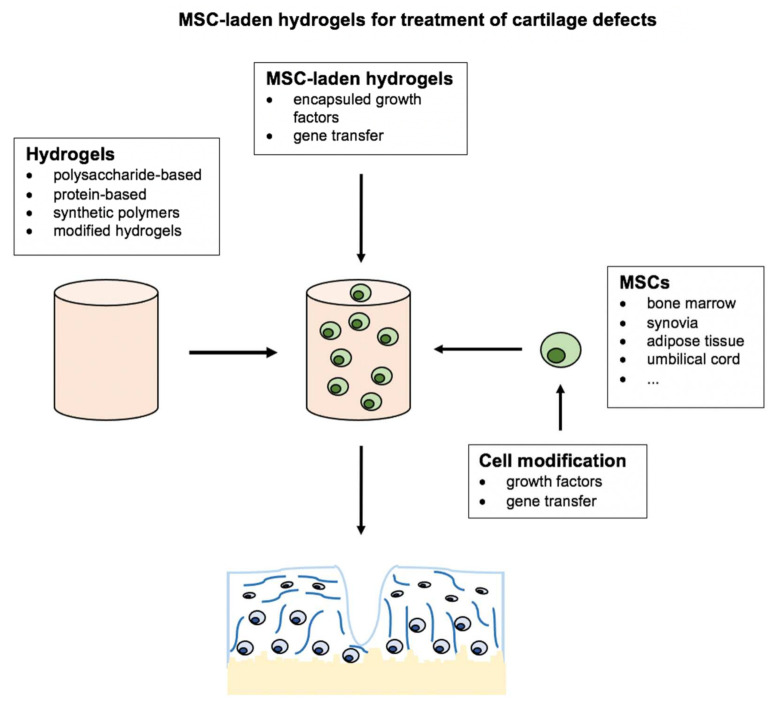
Combinations of hydrogels, mesenchymal stromal cells, and gene transfer, as well as growth factors for cartilage tissue engineering. Image-based on Deng et al., 2020 [14]: Narrative review of the choices of stem cell sources and hydrogels for cartilage tissue engineering.

**Figure 3 gels-07-00217-f003:**
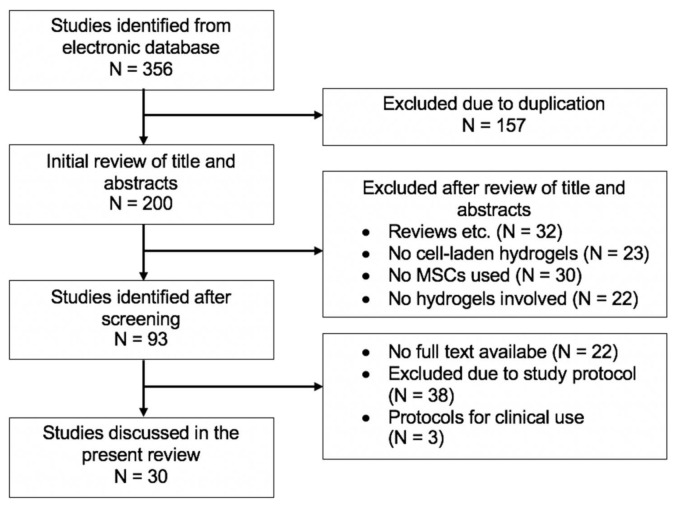
Flowchart for literature research in the present review.

**Table 1 gels-07-00217-t001:** Advantages and disadvantages of selected natural and synthetic hydrogels for cartilage engineering.

Protein-Based Hydrogels	Advantages	Disadvantages
COL/GEL	Collagen as a natural component of the extracellular matrix of hyaline cartilage Immunomodulatory effects Increased cell adhesion Suitable for bioprinting	Poor mechanical properties and stability Temperature-sensitive Limited delivery of growth factors
**Polysaccharide-based hydrogels**	**Advantages**	**Disadvantages**
HA	Natural component of the extracellular matrix of hyaline cartilage Enhances cell functionality and expression of chondrogenic marker genes	Low cell adhesive capacity
AL	Strong mechanical properties Suitable for bioprinting Suitable for gene delivery	Low biodegradability Immunogenic response
AG	Mechanical stability Good viscoelasticity	Limited support of cell functionality
CH	Suitable for delivery of growth factors Similar structure as glycosaminoglycans Highly biodegradable and biocompatible	Limited solubility
**Synthetic hydrogels**	**Advantages**	**Disadvantages**
PEG, PVA	Highly tunable biocompatibility Mechanical properties Partly promote chondrogenesis in seeded cells Mechanical properties	Biologically inert Limited cell adhesive capacity High costs Limited in vivo studies

Precise description of the advantages and disadvantages presented by polysaccharide- and protein-based natural hydrogels as well as synthetic hydrogels. AL: alginate; AG: agarose; CH: chitosan; COL: collagen; GEL: gelatin; HA: hyaluronic acid; PEG: polyethylene glycol; PVA: polyvinyl alcohol.

**Table 2 gels-07-00217-t002:** Preclinical studies examining the use of MSC-laden hydrogels for cartilage engineering.

Author	Year	Animal; Defect Type	Cell Source	Hydrogel	Study Setup	Results
Kim et al. [23]	2011	Rabbits osteochondral defects	BMSCs	HA hydrogel	BMSCs + HA injection BMSCs + 2 HA injections BMSC injection HA injections No treatment	Macroscopic and histological examination with significantly improved healing of osteochondral defects compared to untreated groups Superior histological results for BMSC + 2 HA injections
Lee et al. [75]	2007	Minipigs femoral osteochondral defects	BMSCs	HA hydrogel	BMSCs + hydrogels Hydrogels Saline	Superior histological and macroscopic results for cartilage repair when using BMSC-laden hydrogels compared to control groups
McIlwraith et al. [86]	2011	Horses osteochondral defects femorotibial joint	BMSCs	HA hydrogel	Microfracture + BMSC-laden HA hydrogels Microfracture + HA hydrogels	No clinical or histological differences Macroscopic and arthroscopic better tissue quality and higher aggrecan levels when adding BMSCs
Saw et al. [74]	2009	Goats osteochondral defects	BMSCs	HA hydrogel	Microfracture + BMSCs + 3 HA injections Microfracture + 3 HA injections No treatment	Successful cartilage repair with superior results in groups treated with HA injections and BMSCs
Chiang et al. [77]	2016	Rabbits OA	BMSCs	HA hydrogels	BMSCs + hydrogels Hydrogels No treatment	Improved histological cartilage scores and tissue content in BMSCs + hydrogels group In addition, less cartilage loss and surface abrasion compared to other groups
Mokbel et al. [80]	2011	Donkeys OA	BMSCs	HA hydrogels	BMSCs + hydrogels Hydrogels	Successful repair of cartilage defects in clinical and radiological evaluations in BMSC + hydrogel group compared with control groups
Sato et al. [79]	2012	Pigs OA	BMSCs	HA hydrogels	BMSCs + hydrogels BMSCs Hydrogels Saline	Histological repair of cartilage defects only in group treated with the combination of BMSCs + hydrogels Elevated collagen type II content in the group treated with BMSCs + hydrogels
Desando et al. [76]	2018	Rabbits OA	BMSCs	HA hydrogels	BMSCs BMSCs + hydrogels Bone marrow concentrate Bone marrow concentrate + hydrogels	Successful joint repair evident in all groups HA hydrogels enhance the migration of seeded cells to cartilage BMSCs favor migration to meniscal tissue Bone marrow concentrate favors migration to cartilage
Suhaeb et al. [78]	2012	Rats OA	BMSCs	HA hydrogels	BMSCs + hydrogels BMSCs Hydrogels	Better counteraction of OA progression when using Hydrogels and BMSCs alone compared to the combination of BMSCs + hydrogels
Pascual-Garrido et al. [83]	2019	Rabbits Critical chondral defect in knee trochlea	BMSCs	Novel photopoly-merizable hydrogel	BMSCs + hydrogels Hydrogels Untreated controls	Successful chondrogenesis of seeded BMSCs in vitro Partial cartilage repair in rabbit models in vivo with enhanced chondrogenic differentiation in BMSCs seeded in hydrogels
Critchley et al. [42]	2019	Rabbits Osteochondral defects knee	BMSCs	AL hydrogel	BMSCs + hydrogels Untreated controls	Enhanced repair of cartilage defects with mechanically stable repair tissue in the group treated with BMSCs + hydrogels
Choi et al. [84]	2018	Rabbits Femoral osteochondral defects	BMSCs	GEL hydrogel	Hydrogels BMSCs + hydrogels BMSCS treated with resveratrol + hydrogels	Greater chondrogenic potential in BMSCs treated with resveratrol Increased expression of chondrogenic marker genes and decreased expression of hypertrophic marker genes in BMSCs treated with resveratrol
Kim et al. [85]	2014	Rats OA	BMSCs	SAP hydrogels	BMSC + hydrogels BMSCs Hydrogels Negative controls	Anti-inflammatory and chondroprotective effects, decrease in apoptosis markers in the group treated with BMSCs + hydrogels Increased bone mineral density in the group treated with BMSCs + hydrogels

AL: alginate; BMSCs: bone marrow-derived MSCs; GEL: gelatin; HA: hyaluronic acid; OA: Osteoarthritis; SAP: self-assembled peptide.

**Table 3 gels-07-00217-t003:** Clinical studies examining the use of MSC-laden hydrogels for cartilage engineering.

Author	Year	Defect Type	Cell Source	Hydrogel	Study Design	Results
Leet et al. [82]	2012	Chondral lesion	BMSCs	HA hydrogels	BMSCs + hydrogel Microfracture + hydrogel	Less-invasive surgery method for application of BMSCs + hydrogel No significant differences regarding the functional outcome and patient satisfaction
Pipino et al. [87]	2019	Osteochondral defect	ADMSCs	Polyglucosamine/glucosamine carbonate hydrogels	Microfractures + ADMSCs + hydrogel Microfracture	High patient satisfaction following treatment with ADMSCs + hydrogels Enhanced cartilage repair in the group treated with ADMSCs + hydrogels

ADMSCs: adipose-derived mesenchymal stromal cells; BMSCs: bone marrow-derived MSCs; HA: hyaluronic acid.

**Table 4 gels-07-00217-t004:** Preclinical studies examining the use of different combinations of MSC-subpopulations and hydrogels for cartilage engineering.

Author	Year	Animal; Defect Type	Cell Source	Hydrogel	Study Design	Results
Lv et al. [88]	2018	Sheep OA	ADMSCs	HA hydrogel	ADMSCs (high dose) + hydrogel ADMSCs (low dose) + hydrogel Stromal vascular fraction Hydrogel Saline	Superior results in the delay of OA progression and cartilage repair when using ADMSCs and hydrogels
Feng et al. [89]	2018	Sheep OA	ADMSCs	HA hydrogels	ADMSCs (high dose) + hydrogel ADMSCs (low dose) + hydrogel Hydrogel Saline	Anti-inflammatory effects and repair tissue with typical articular cartilage features when using ADMSCs and hydrogels
Sevastianov et al. [90]	2021	Rabbit Knee OA	ADMSCs	COL hydrogels	ADMSCs + hydrogels ADMSCs + decellularized porcine articular cartilage	Cell-laden hydrogels were superior in stimulating cartilage repair in vivo MSCs cultured in the presence of decellularized porcine articular cartilage formed more cartilage-specific ECM components
Jia et al. [91]	2019	Rabbits Osteochondral defects of the knee	Synovial fluid-derived MSCs	CH hydrogels	MSCs + hydrogel Hydrogel Untreated controls	Improved macroscopic appearance and histological results in the group treated with MSCs + hydrogels
Wu et al. [92]	2019	Minipigs Osteochondral defects	Umbilical cord-derived MSCs	HA hydrogels	MSCs + hydrogel Untreated controls	Effective treatment of cartilage defects using MSCs + hydrogels in minipigs
Li et al., [93]	2018	Rats Osteochondral defect of the knee	Arthroscopic flushing fluid-derived MSCs	Polypegda/HA hydrogels	MSCs + hydrogel Hydrogel Untreated controls	Significantly improved

ADMSCs: adipose-derived mesenchymal stromal cells; CH: chitosan; HA: hyaluronic acid; OA: osteoarthritis.

**Table 5 gels-07-00217-t005:** Studies examining the combined use of MSCs, growth factors/gene transfer, and hydrogels for cartilage engineering.

Author	Year	Animal; Defect Type	Cell Source; Hydrogel	Growth Factors; Gene Transfer	Study Design	Results
Vayas et al. [96]	2021	Rabbits Chondral defect	ADMSCs PLGA hydrogel	BMP-2	Microfracture BMSCs + hydrogel + BMP-2 BMSCs + hydrogel Untreated defects	Significantly enhanced cartilage repair in all groups except untreated controls and microfracture treatment alone
Deng et al. [97]	2019	Mice Intramuscular implantation	BMSCs PDLLA-PEG/HA hydrogel	TGF-ß3	BMSCs + PDLLA-PEG hydrogel + TGF-ß3 BMSCs + PDLLA-PEG/HA hydrogel + TGF-ß3	Controlled release of TGF-ß3 after addition of HA to hydrogels Higher GAG production, higher mechanical strength, and increased chondrogenic gene expression after addition of HA to hydrogels
Sathy et al. [43]	2019	Mice Subcutaneous implantation	BMSCs AL hydrogel	DMOG TGF-ß3 BMP-2	BMSCs + hydrogel + DMOG + TGF-ß3 + BMP-2 BMSCs + hydrogel + TGF-ß3 + BMP-2	Enhanced chondrogenesis and reduced hypertrophy after addition of hypoxia-mimicking factor DMOG
Jooybar et al. [98]	2019	In vitro study	BMSCs HA-tyramine hydrogel	Platelet lysate	MSCs + hydrogel + platelet lysate MSCs + hydrogel	Enhanced production of collagen type II and proteoglycans and a tough, dense matrix in hydrogels with platelet lysate
Xia et al. [99]	2009	Mice Subcutaneous implantation	BMSCs PGA hydrogel	Adenoviral gene transfer of *TGFB1*	MSCs + hydrogel + adenoviral gene transfer of *TGFB1* MSCs + hydrogel	Successful neocartilage formation after subcutaneous implantation in mice Significantly increased GAG content and production of collagen type II
Wang et al. [100]	2014	Pig Knee cartilage defect	BMSCs Demineralized bone matrix	Adenoviral gene transfer of *TGFB3* and *BMP2*	MSCs + hydrogel + adenoviral gene transfer of *TGFB3* and *BMP2* BMSCs + hydrogel + adenoviral gene transfer of *TGFB3* BMSCs + hydrogel + adenoviral gene transfer of *BMP2* BMSCs + hydrogel	Enhanced histological cartilage repair in the group with adenoviral gene transfer of *TGFB3* and *BMP2*
Cao et al. [101]	2011	Rabbit Cartilage defect	BMSCs PGA hydrogel	Adenoviral gene transfer of *SOX9*	BMSCs + hydrogel + adenoviral gene transfer of *SOX9* BMSCs + hydrogel	Repair of full-thickness cartilage defects in rabbit models Enhanced repair, more cartilage-like tissue, and cartilage-like ECM in the group treated with adenoviral gene transfer of *SOX9*
Weißenberger et al. [19]	2020	In vitro study	BMSCs COL type I hydrogel	Adenoviral gene transfer of *SOX9, TGFB1,* or *BMP2*	BMSCs + hydrogel + adenoviral gene transfer of *SOX9* BMSCs + hydrogel + adenoviral gene transfer of *TGFB1* BMSCs + hydrogel + adenoviral gene transfer of *BMP2* BMSCs + hydrogel	Enhanced chondrogenic differentiation and decreased hypertrophic de-differentiation in BMSC-laden hydrogels treated with adenoviral gene transfer of *SOX9*
Venkatesan et al. [102]	2018	In vitro study	BMSCs Fibrin/ polyurethane hydrogel	Recombinant adeno-associated gene transfer of *SOX9*	BMSCs + hydrogel + recombinant adeno-associated gene transfer of *SOX9* BMSCs + hydrogel	Enhanced chondrogenesis in cultures treated with recombinant adeno-associated gene transfer of *SOX9*
Lu et al. [103]	2014	Rabbits Chondral defects	ADMSCs PLGA hydrogel	Baculoviral gene transfer of *TGFB3* and *BMP6*	ADMSCs + hydrogel + Baculoviral gene transfer of *TGFB3* and *BMP6* (short and prolonged expression) ADMSCs + hydrogel	Prolonged expression of *TGFB3* and *BMP6* led to the successful repair of full-thickness cartilage defects with native matrix composition, mechanical structure, and zonal formation
Lee et al. [104]	2012	Rat Chondral defects	ADMSCs Fibrin hydrogel	Retroviral gene transfer of *SOX5, SOX6,* and *SOX9*	ADMSCs + hydrogel + recombinant adeno-associated gene transfer Retroviral gene transfer of *SOX5, SOX6, SOX9,* or of the SOX-trio ADMSCs + hydrogel + treatment with TGF-ß2 and BMP-7	Successful repair of full-thickness cartilage defects in vivo in all groups Significantly enhanced expression of chondrogenic marker genes, production of collagen type II, and GAG content in the group transduced with the SOX-trio

ADMSCs: adipose-derived mesenchymal stromal cells; BMP: bone morphogenetic protein; BMSCs: bone marrow-derived MSCs; COL: collagen; DMOG: dimethyloxalylglycine; GAGs: glycosaminoglycans; HA: hyaluronic acid; IGF-1: insulin-like growth factor 1; PDLLA-PEG: poly-dl-lactic acid/polyethylene glycol/poly-dl-lactic acid; PGA; poly(glycolic acid); PLGA: Poly(lactide-co-glycoside); SOX: SRY-Box Transcription Factor; TGF: transforming growth factor.

## Data Availability

The datasets used and analyzed during the current study are available from the corresponding author on reasonable request.

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
