# Peer review of "Combinations of Hydrogels and Mesenchymal Stromal Cells (MSCs) for Cartilage Tissue Engineering—A Review of the Literature"

_gels, 2021, doi:10.3390/gels7040217_

Round 1
Reviewer 1 Report
The review, which has an illustrative generalizing character, is devoted to the subject quite popular recently related to the use of MSC-laden hydrogels for cartilage regeneration, which is discussed in a significant number of original articles, reviews and monographs.
Comments and questions
- The depth of literature research is not specified.
- I also advise you to pay attention to the following publications: doi 10.1039/c6bm00863a, doi 10.1007/978-3-319-40144-7, doi 10.1007/978-3-319-51617-2 and doi.org/10.3390/life11080756 (MDPI journal).
- Question: How original is Figure 1, which I saw earlier? If it is not from the publication of the authors of the review, then at least a link to the corresponding publication is necessary.
- I advise the authors to check the manuscript more carefully for grammatical and technical errors, typos, for example, among others, see lines 90, 100, 101, 112, 151, 208, 230, 274-275, 277, 288-289, 305-306, Conclusion.
Author Response
We thank the reviewer for his remarks. We replied to all comments in the following word document.

Reviewer 2 Report
Dear authors,
Your review "Combinations of hydrogels and mesenchymal stromal cells (MSCs) for cartilage tissue engineering - a review of the literature" is indeed of scientific interest and certainly deserves space in the "gels" journal. The aim of this review is to add to the understanding of disadvantages and advantages of various combinations of MSC-subpopulations, growth factors, gene transfers and hydrogels in cartilage engineering. Unfortunately, despite the scientific interest of your review, some arrangements must be made before publication.
1) There are two part of Materials and methods and Results sections in the review, please merge the two parts according to the requirements of "gels" journal.
2) The alignments of text and bullets in the Table 2, Table 3 and Table 4 are messy, they need minor changes to achieve the effect of Table 1.
3) In the section “A clinical trial examined the use of BMSC-laden HA hydrogels for the treatment of osteochondral defects of the knee and found significant improvements in functionality and pain reduction comparable to patients treated with microfracture alone (Table 2)” (on line 334), authors want to summarize BMSC-laden HA hydrogels for the treatment of osteochondral defects in clinical trial. But the content shown in the Table 2 is the application of BMSC and hydrogel, not just HA hydrogel in preclinical research. Please correct this contradiction
Author Response

(The authors gave the same response as above.)

Round 2
Reviewer 1 Report
All comments are taken into account by the authors.